

**Field measurements of methylglyoxal using Proton Transfer**
**Reaction-Time of Flight Mass Spectrometry and comparison**
**to the DNPH/HPLC-UV method**
Vincent Michoud[1,2], Stéphane Sauvage[1], Thierry Léonardis[1], Isabelle Fronval[1], Alexandre
Kukui[3], Nadine Locoge[1], Sébastien Dusanter[1]
[1] IMT Lille Douai, Univ. Lille, SAGE - Département Sciences de l'Atmosphère et Génie de
l'Environnement, 59000 Lille, France
[2] LISA/IPSL, Laboratoire Interuniversitaire des Systèmes Atmosphériques, UMR CNRS 7583,
Université Paris Est Créteil (UPEC) et Université Paris Diderot (UPD), Créteil, France
[3] Laboratoire de Physique et Chimie de l'Environnement et de l'Espace (LPC2E), UMR6115 CNRS-
Université d'Orléans, 45071 Orléans CEDEX 2, France
**Abstract**
Methylglyoxal (MGLY) is an important atmospheric α-dicarbonyl species whose
photolysis acts as a significant source of peroxy radicals, contributing to the oxidizing capacity
of the atmosphere and, as such, the formation of secondary pollutants such as organic aerosols
and ozone. However, despite its importance, only a few techniques exhibit time resolutions and
detection limits that are suitable for atmospheric measurements.
This study present, to the best of our knowledge, the first measurements of ambient
MGLY using Proton Transfer Reaction-Time of Flight Mass Spectrometry (PTR-ToFMS).
These measurements were performed during the ChArMEx SOP2 field campaign. This
campaign took place at a Mediterranean site characterized by intense biogenic emissions and
low levels of anthropogenic trace gases. Concomitant measurements of MGLY were performed
using the 2,4-dinitrophenylhydrazine (DNPH) derivatization technique and High Performance
Liquide Chromatography (HPLC) with UV detection. PTR-ToFMS and DNPH-HPLC
measurements were compared to determine whether these techniques can perform reliable
measurements of MGLY.
Ambient time series revealed levels of MGLY ranging from 28-365 pptv, with a clear
diurnal cycle due to elevated concentrations of primary biogenic species during daytime, whose
oxidation led to large production rates of MGLY. A scatter plot of the PTR-ToFMS and DNPH-
HPLC measurements indicates a reasonable correlation ($R^2$=0.48) but a slope significantly





lower than unity (0.58±0.05) and a significant intercept of 88.3±8.0 pptv. A careful
investigation of the differences between the two techniques suggests that this disagreement is
not due to spectrometric interferences from $H_3O^+(H_2O)_3$, MEK (or butanal) detected at m/z
73.050 and m/z 73.065, respectively, which are close to the MGLY m/z of 73.029. The
differences are more likely due to uncorrected sampling artefacts such as overestimated
collection efficiency or loss of MGLY into the sampling line for the DNPH-HPLC technique
or unknown isobaric interfering compounds such as acrylic acid and propanediol for the PTR-
ToFMS.
Calculations of MGLY loss rates with respect to OH-oxidation and direct photolysis
indicate similar contributions for these two loss pathways.
**1 Introduction**
Methylglyoxal (MGLY, $CH_3C(O)CHO$) is an important α-dicarbonyl species in the
atmosphere. It is mainly produced during the oxidation of Volatile Organic Compounds (VOCs)
amongst which isoprene and acetone are the main contributors. Fu et al., (2008) calculated
production rates of 110 and 10 Tg year$^{-1}$ from the oxidation of isoprene and acetone,
respectively. Other precursors of MGLY are $C_3$-$C_5$ isoalkanes (Jacob et al., 2002), aromatic
compounds (Volkamer et al., 2001; Pan and Wang, 2014; Wu et al., 2014)), and monoterpenes
(Fick et al., 2003; Nunes et al., 2005). Due to the anthropogenic and biogenic natures of MGLY
precursors, this compound can therefore be found in significant levels (low tens to hundreds of
pptv) in urban, rural or even remote and marine environments (Henry et al., 2012 and references
therein).
The principal sink of MGLY is thought to be photolysis (Fu et al., 2008), which can
significantly contribute to the formation of $RO_x$ ($OH+HO_2+RO_2$) radicals in the troposphere
(Dusanter et al., 2009), which in turn can enhance the formation rates of secondary pollutants,
including ozone and Secondary Organic Aerosol (SOA). In addition, MGLY has been identified
as a direct precursor of SOA (Altieri et al., 2008; Hallquist et al., 2009), due to aqueous
reactions in clouds leading to the formation of oligomers and oxalic acids, which can then form
SOA upon cloud droplet evaporation (Altieri et al., 2008).
Despite the important role of MGLY in the atmosphere, there are only a few measurement
techniques, most of them being expensive, requiring highly skilled operators, or suffering from
low time resolution. A common method relies on chemical derivatization and chromatographic





analysis. Several derivatization agents can be used to trap carbonyl compounds such as 2,4-
dinitrophenylhydrazine (DNPH) (Lee et al., 1998; Ho et al., 2014a; Lawson et al., 2015), o-(2,
3, 4, 5, 6-pentafluorobenzyl)hydroxylamine (PFBHA) (Spaulding et al., 1999; Ho and Yu,
2002; Ortiz et al., 2006, 2013; Temime et al., 2007) and pentafluorophenylhydrazine (PFPH)
(Ho and Yu, 2004; Pang and Lewis, 2011; Pang et al., 2011; Dai et al., 2012). Methods relying
on chemical derivatization imply active sampling through cartridges or liquid solutions
containing the selected reagent and a subsequent offline analysis using Gas Chromatography-
Mass Spectrometry (GC-MS) or High Performance Liquid Chromatography with ultraviolet
detection (HPLC-UV). Low detection limits are reached for these techniques with the
advantages of monitoring several carbonyl compounds simultaneously. Indeed, Ho and Yu
(2004) reported detection limits below 0.3 ppbv for a large range of carbonyl compounds,
including formaldehyde, acetaldehyde, propanal, acrolein, glyoxal, MGLY and others. These
authors used cartridges loaded with PFPH on a Tenax sorbent, a sampling time of 4 h and a
sampling flow rate of 100 mL min$^{-1}$. Ait-Helal et al. (2014) even reported lower detection limits
ranging from 10-60 pptv for C1-C9 aldehydes and ketones, including MGLY, using a sampling
duration of 3h for DNPH cartridges and a sampling flow rate of 1.5 L min$^{-1}$. The cartridges
were analysed by HPLC-UV. However, the main drawback of these methods is the low time
resolution of typically 3-4 h, which is too long to investigate photochemical processes.

19       Alternative techniques based on mist chambers and derivatization solutions such as

PFBHA (Spaulding et al., 2002) or DNPH (Munger et al., 1995) were used to measure MGLY
with a faster time resolution of approximately 10 min and low limits of detection (LOD).
Spaulding et al. (2002) reported a LOD of 7.7 pptv at a sampling flow rate of 25-70 L min$^{-1}$
(Spaulding et al., 2002). More recently, a microfluidic derivatization approach using PFBHA
and a planar glass micro-reactor was developed to measure glyoxal and MGLY at a time
resolution of 30 min and sampling flow rate ranging from 100 to 600 mL min$^{-1}$ (Pang et al.,
2014). This setup exhibits LODs of 76 and 185 pptv (3σ) for glyoxal and MGLY, respectively.
The authors also report the use of a Solid Phase MicroExtraction (SPME) method, previously
described by Gomez Alvarez et al. (2012), capable of measuring MGLY with a LOD of
150 pptv (3σ) and a measurement time of 25 min. The SPME technique relies on a derivatization
of aldehyde species into oximes on a fibre loaded with PFBHA and a subsequent analysis by
Gas Chromatography-Flame Ionization Detection (GC-FID). Gomez Alvarez et al. (2007) also
mentioned the possibility to measure MGLY using a SPME instrument as well as a Gas
Chromatography-Electron Capture Detector (GC-ECD), both calibrated against Fourier
Transform Infrared Spectroscopy (FTIR).



In addition to these chemical derivatization methods, optical/spectroscopic approaches
have also been employed to measure MGLY. Henry et al. (2012) reported a Laser-Induced
Phosphorescence (LIP) instrument capable of simultaneous measurements of glyoxal and
MGLY with a time resolution of 5 min and LODs of 4.4 and 243 pptv (3σ), respectively.
Thalman and Volkamer (2010) developed a blue LED (Light-Emitting Diodes) Cavity
Enhanced Differential Optical Absorption Spectroscopy (CE-DOAS) instrument for in-situ
measurements of MGLY among other compounds (nitrogen dioxide, glyoxal, iodine oxide and
water vapour). This instrument exhibits a LOD of 170 pptv (2σ) at a time resolution of 1 min.
Incoherent Broadband Cavity Enhanced Absorption Spectroscopy (IBBCEAS) has also been
used to measure both Glyoxal (Washenfelder et al., 2011) and MGLY (Pang et al., 2014), with
a LOD of 1 ppbv (3σ) for a measurement time of 20 s for the latter. FTIR is another
spectroscopic method capable of measuring MGLY (Talukdar et al., 2011). However, FTIR
exhibits a LOD in the ppbv range, which is not low enough for ambient measurements, even
with a long path length of hundreds of meters (Pang et al., 2014). Overall, while these
spectroscopic techniques usually exhibit performances that are suitable for atmospheric
measurements, they also require highly skilled operators and the use of fragile pieces of
equipment (light sources, mirrors etc…).
The use of Proton Transfer Reaction-Time of Flight Mass Spectrometry (PTR-ToFMS)
has been attempted for Glyoxal measurements by Stonner et al. (2017). However, these authors
showed that the sensitivity of PTR-ToFMS instruments was too low to monitor ambient
concentrations. MGLY measurements by PTR-ToFMS have been reported by Pang et al. (2014)
and Thalman et al. (2015) during intercomparison experiments. Pang et al. (2014) observed a
significant disagreement between PTR-ToFMS measurements and results from other
techniques (Microfluidic derivatization, IBBCEAS, FTIR, SPME) during photo-oxidation
experiments of isoprene under low $NO_x$ conditions in the EUPHORE chamber. According to
the authors, this disagreement was due to interferences from $(H_2O)_3.H_3O^+$ at m/z 73 (no
deconvolution of peaks within this mass unit). Thalman et al. (2015) also reported interferences
from the $(H_2O)_3.H_3O^+$ cluster and the fragmentation of larger compounds upon protonation.
However, blank measurements made at the same relative humidity than in ambient air should
contain the contribution of $(H_2O)_3.H_3O^+$ and frequent blank measurements, as usually done
during field campaigns, could easily be subtracted to reduce the impact of $(H_2O)_3.H_3O^+$ on the
MGLY measurements. De Gouw and Warneke (2007) reported measurements of
methylethylketone (MEK) at the same unit mass using a PTR-MS equipped with a quadrupole.
However, Time of Flight mass spectrometers provide the opportunity to deconvolve signals of





MGLY (m/z 73.029) and MEK (m/z 73.065), which are separated by 0.036 Daltons. Thus, if the mass resolution of the PTR-ToFMS instrument is sufficient, an adequate peak fitting procedure and frequent blank measurements should allow a selective detection of methylglyoxal.

In this study, we present online measurements of MGLY using Proton Transfer Reaction-Time of Flight Mass Spectrometry (PTR-ToFMS). This study describes a procedure to conduct measurements of MGLY using PTR-ToFMS, reports a comparison of PTR-ToFMS and DNPH-HPLC measurements performed during an intensive field campaign in the Mediterranean basin, and presents an investigation of the MGLY loss rate during this campaign.

**2 Experimental**

**2.1 The Chemistry-Aerosol Mediterranean Experiment (ChArMEx)**

The ChArMEx SOP2 (Short Observation Period 2) field campaign took place from 15 July to 05 August at Cape Corsica (France) on a hilltop (alt. 533 m) within a wind farm (42.969°N, 9.380°E). It is a coastal site surrounded by the sea a few km away in all directions (2.5-6 km) (Zannoni et al., 2015). The site was covered by typical Mediterranean vegetation ("maquis" shrub-land) (Zannoni et al., 2015) leading to large emissions of biogenic VOCs and elevated concentrations of isoprene (up to 1.3 ppbv) and monoterpenes (up to 2.2 ppbv) (Michoud et al., 2017). Since MGLY is a byproduct of isoprene oxidation ($2^{nd}$ & $3^{rd}$ generations), this site is of interest to perform and investigate its budget. On the contrary, low anthropogenic influence was observed at the measurement site since the closest city, Bastia, is located ~30 km away (Michoud et al., 2017).

**2.2 PTR-ToFMS measurements**

Measurements of MGLY, among other species (Michoud et al., 2017), were conducted using a PTR-ToFMS instrument from KORE Inc™ ($2^{nd}$ generation). Ambient air was sampled through a 5-m long line made of PFA (PerFluroAlkoxy). The line was held at 50°C and the flow rate was set at 1.2 L min$^{-1}$ to reduce the residence time below 4-s. The PTR-ToFMS sampled from this line at a constant flow rate of 150 mL min$^{-1}$. Reactor pressure and temperature were set at 1.33 mbar and 40°C, respectively, leading to an E/N value of 135 Td. The PTR-ToFMS spectra were integrated over 10 minutes, leading to 6 measurements per hour.



An automatized zero procedure was performed for 10 minutes every hour to subtract
potential contaminations from the lines and to suppress interferences from water clusters and
other ions formed inside the glow discharge. Zero air was generated by passing ambient air
through a catalytic converter (1/2" stainless steel tubing filled with 2 grams of Pt wool held at
350°C) allowing to zero the instrument at the same relative humidity than in ambient air. In
order to test the efficiency of the catalytic converter, mixtures of several tens of hydrocarbons
at the ppb level were passed through the converter and the remaining VOCs were measured by
GC analyzers. Levels lower than the detection limits of the GCs (5-10 pptv) were observed,
indicating an efficient removal of the VOCs.
VOC signals were extracted from the 10-min mass spectra by summing the number of
counts detected within m/z windows centred on the exact masses of the VOCs of interest
(m/z$_{VOC}$ ± 0.21). These signals were normalized by the signals of $H_3O^+$ and the ionic water
cluster $H_3O^+(H_2O)$ as proposed by de Gouw and Warneke (2007). VOC concentrations were
then calculated using Eq. 1.

$$[RH] = \frac{i_{RH\_net}}{(i_{H_3O^+} \times 500 + X_r . i_{H_3O^+(H_2O)} \times 250)} \cdot \frac{150000}{R_f} \qquad (1)$$

Where i$_{RH\_net}$ is the net VOC signal (difference of signals recorded when sampling
ambient and zero air), $i_{H_3O^+}$ the signal from $H_3O^+$ ions at m/z 21, $i_{H_3O^+(H_2O)}$ the signal from
$H_3O^+(H_2O)$ at m/z 39, $X_r$ a factor to account for the effect of humidity on the PTR-ToFMS
sensitivity (de Gouw and Warneke, 2007), $R_f$ the sensitivity determined by calibration (in
ncps ppb$^{-1}$) and 150000 the corresponding number of primary $H_3O^+$ ions in the PTR-ToFMS
reactor (in cps). The instrument was calibrated every three days during the campaign using a
Gas Calibration Unit (IONICON®) and various standards (RESTEK, PRAXAIR) made of
hydrocarbons (isoprene, benzene, toluene, o-xylene, ethylbenzene, α-pinene) and mono-
functional oxygenated VOCs (methanol, acetaldehyde, acetone, methylethylketone). These
calibrations were performed at a relative humidity of 50% at 20°C without passing by the entire
5-m long heated sampling line. $X_R$ was determined by conducting additional calibrations at
various relative humidity values before and after the campaign. The calibration factor, $R_f$ in Eq.
1, was normalized to 150000 cps of reagent ions. Specific calibrations performed for
methylglyoxal are described in section 3.1.
As mentioned in the introduction, MGLY and MEK are detected at m/z 73.029 and
73.065, respectively. A Gaussian peak fitting operation was performed to deconvolve the two
peaks observed in the m/z window 72.95-73.15 during ambient sampling, using the curvefit





tools from Grams™ software (Thermo Scientific™) (see supplementary material figure S1). The
signals recorded in this window were accumulated over 1 h to reduce the time needed for this
procedure, which was made manually. An automatic peak fitting operation is planned in the
future via the development of a software. The MEK-to-methylglyoxal ratio of areas observed
for the 1 h cumulated signals was then applied to each 10 min recorded signals (total number
of counts recorded within the m/z window 72.95-73.15) providing measurements of
methylglyoxal and MEK at a 10 min time resolution. It is worth noting that the MGLY lifetime
of at least 1 h and the longer lifetime of MEK ensure that the MGLY-to-MEK ratio does not
change significantly over an hour. Once the signals were deconvolved for each compound, the
procedure described in the previous paragraph was applied to calculate their ambient
concentrations using appropriate sensitivity and humidity dependence factors.

12        The $3\sigma$ detection limits were calculated from the hourly blank measurements. The

average detection limit for methylglyoxal during the whole campaign is 22 pptv ($3\sigma$) at the time
resolution of 10 min. The total uncertainty was estimated following the "Aerosols, Clouds, and
Trace gases Research InfraStructure network" guidelines (ACTRIS Measurement Guideline
VOC, 2012), taking into account precision and systematic errors. The repeatability on MGLY
measurements was calculated as the square root of the net signal ($i_{RH\_net}$) since the statistic for
PTR-ToFMS signals follows a Poisson distribution (de Gouw and Warneke, 2007) and was on
average 9±3%. The systematic errors concerned the calibration factor ($R_f$) and the peak fitting
procedure and are estimated to be 22% for methylglyoxal.
**2.3 Active sampling on DNPH cartridges**

24        Measurements of carbonyl compounds from $C_1$ to $C_8$, including MGLY and MEK, were

performed using DNPH cartridges (Waters™) and an automatic sampler (ACROSS-TERA
Environment™), based on the US EPA TO-11A method. The analysis of the cartridges was
performed in the laboratory using HPLC-UV (Waters 2695 & 2487). This deployment has
already been described by Ait-Helal et al. (2014) and Michoud et al. (2017). Ambient air was
sampled through a 3-m long PFA line (1/4") at a height of 1.5 m above the roof of the trailer
next to the PTR-ToFMS sampling line. This air was collected for 3 h on each cartridge at a flow
rate of 1.5 L min$^{-1}$. A potassium iodide (KI) ozone scrubber and a stainless steel particle filter
(porosity: 2µm) were setup on the sampling line before the automatic sampler. The $3\sigma$ detection
limit was determined to be 6 pptv for MGLY from blank cartridges (unused cartridges stored





under similar conditions than exposed cartridges). The systematic error is estimated to be 25%
for these measurements.
The HPLC-UV instrument used to analyse the DNPH samples was calibrated using a
standard solution of hydrazone compounds (TO11/IP-6A) commercialized by SUPELCO.
However, MGLY-DNPH is not present in this solution and a hydrazone standard was made by
mixing a known volume of an aqueous solution of MGLY (40% in water, Acros Organics™)
into an excess of acidified DNPH solution. It is worth noting that calibrating the HPLC-UV
using a liquid standard of hydrazones is based on the assumption that the collection efficiency
of carbonyl compounds through DNPH cartridges is 100%.
**2.4 Investigation of the Methylglyoxal loss rate**
Two sinks were considered in the steady state loss calculations: reaction of MGLY with
OH and MGLY photolysis. The loss from the reaction with OH was calculated using
concentrations of both MGLY and OH, the latter being measured by Chemical Ionisation Mass
Spectrometry (Kukui et al. 2008), and the recommended rate constant of $1.50 \times 10^{11}$ cm$^3$
molecule$^{-1}$ s$^{-1}$ (Atkinson et al., 2006). $J(NO_2)$ and the photolysis frequencies for some other
species were derived from the actinic flux measured with an actinic flux spectroradiometer
METCON 6007 (Meteorologie Consult GmbH). However, Photolysis frequencies for MGLY
were not derived from these measurements. The approach described in Dusanter et al. (2009)
was therefore employed to calculate $J(MGLY)$ and $J(NO_2)$ as a function of the solar zenith
angle for the measurement site (lat: 42.969°N, long: 9.380°E) using the Master Chemical
Mechanism (MCM) parameterization (Jenkin et al. 1997; Saunders et al., 2003). This
parameterization was derived for an ozone column of 345 Dobson, an altitude of 500 m and
clear sky conditions. Calculated values of $J(MGLY)$ were then corrected for differences in
altitude, cloud covering, aerosol and $O_3$ column densities using a scaling factor derived from
the measured-to-calculated $J(NO_2)$ ratio. The photolytic loss of MGLY was calculated using
these scaled photolysis frequencies and PTR-ToFMS measurements of MGLY.

**3 Results and Discussion**

**3.1 PTR-ToFMS calibrations for MGLY**



These calibrations were not performed during the field campaign but a few months later
using a Liquid Calibration Unit (LCU, IONICON™) and an aqueous solution of MGLY (40%,
Acros Organics™) (see Figure 1). The LCU allows generating a standard mixture containing the
targeted compounds at known mixing ratios by evaporating an aqueous solution of these
compounds into a large flow of zero air (1.0 L min$^{-1}$ in our case). The standard solution flows
was varied between 1 and 20 μL min$^{-1}$ to generate MGLY concentrations ranging from 0.6 to
11 ppbV.

8       Figure 1 shows that the PTR-ToFMS response is linear with the MGLY concentration

over the tested range, with no significant offset. While the lower limit of the tested range is
larger than observed ambient concentrations (0.05-0.3 ppbv, figure 2), it has to be noted that
the PTR-MS response has always been observed to be linear with the analyte concentration and
a linear response is expected for MGLY for mixing ratios below 0.6 ppbv. These calibration
experiments indicate an averaged calibration factor of 2.54±0.49 ncps ppb$^{-1}$ when normalized
to 150000 cps of reagent ions and using a Xr factor set to 0.5. It is worth noting that changing
the flow rate of the liquid standard solution to generate various MGLY concentrations leads to
a change in humidity in the gas exiting the LCU, tracked by the m/z 37-to-m/z 19 ratio (varying
from 0.1 to 0.5) during these calibration experiments. The good linearity observed in Fig. 1
gives confidence in the Xr factor value used to determine the calibration factor. Therefore, the
same Xr value of 0.5 was used for ambient measurements of MGLY.

20       To account for a potential drift in sensitivity between the field measurements and the

calibration experiments performed later in the laboratory, calibrations of MEK were also
performed during the laboratory experiments using a Gas Calibration Unit (GCU, IONICON™)
and a standard mixture provided by IONICON (Restek™). A comparison between MEK
response factors observed during field measurements and laboratory experiments allows
accounting for a drift of the PTRMS sensitivity as further discussed below. The Restek mixture
contains 15 compounds including 0.99±0.05 (2σ) ppmv of MEK. Calibrations of MEK were
performed every 3 days during the field experiments using the GCU and the same Restek
mixture. Since MEK is detected at the same mass unit than MGLY, a change in sensitivity for
MGLY between the field measurements and the laboratory calibrations due to a change in ion
transmission inside the mass spectrometer would also be observed for MEK. A ratio of the
calibration factors measured for MGLY and MEK during the laboratory experiments was used
to calculate the MGLY calibration factor from the calibration factor measured for MEK during
the field campaign. The laboratory calibrations led to an averaged sensitivity factor of
6.60±0.16 ncps ppb$^{-1}$ for MEK when normalized to 150000 cps of reagent ions, leading to a





MGLY-to-MEK sensitivity ratio of 0.38. During the ChArMEx field campaign, an averaged
calibration factor of $7.57\pm0.52$ ncps ppb$^{-1}$ was observed for MEK when normalized to 150000
cps of reagent ions, indicating a decrease of approximately 13% between the field and
laboratory measurements. However, as mentioned above, using the MGLY-to-MEK sensitivity
ratio determined in the laboratory allows correcting for this change.
**3.2 Time series of MGLY**

9         Concomitant measurements of MGLY by PTR-ToFMS and the DNPH/HPLC-UV

method in an environment characterized by intense biogenic emissions represent a good
opportunity to test how the two techniques compare for this compound. Figure 2 presents time
series of MGLY measurements from PTR-ToFMS (red) and active sampling on DNPH
cartridges (black) from 15 July to 6 August. The PTR-ToFMS measurements performed at a
time resolution of 10 min were averaged over 3 h around the sampling middle time of each
cartridge measurement to allow a direct comparison between the two techniques. This figure
shows that significant levels of MGLY were observed during the campaign, with concentrations
ranging from 30 to 370 pptv. In addition, these measurements indicate clear diurnal variations,
which is consistent with similar variations of MGLY precursors of biogenic origin observed
during the campaign, e.g. isoprene and monoterpenes (see Figure 2, middle panel). Figure 3
displays campaign averaged diurnal profiles for MGLY and indicates daily maxima observed
around 13:45 local time (Central European Summer Time +02:00 UTC) when the
photochemistry is the most intense (see Figure 2, bottom panel).
**3.3 Comparison of MGLY measurements**

26         Overall, a reasonable agreement is observed between the two techniques (see Figure 2),

except for 17 July, 25 July and the last 4 days where the measured PTR-ToFMS concentrations
were higher by 16-148%. A close look at Figure 2 also indicates that PTR-ToFMS
measurements are usually higher at night and the concentrations do not decrease as low as that
observed for the cartridges. For example, 3-h averaged PTR-ToFMS concentrations measured
at 1:30, 4:30 and 22:30 (local time) for the overall campaign (Figure 3) are 127, 144 and
136 pptv, respectively, which are approximately 14, 21 and 30 pptv higher (11-22%) than
cartridge measurements, respectively.





Figure 4 displays a scatter plot of PTR-ToFMS vs. DNPH/HPLC-UV measurements.
While a reasonable correlation is found between the two techniques ($R^2$=0.48), a significant
intercept of 88±16 pptv (1σ) confirms the higher concentrations observed by PTR-ToFMS at
night, suggesting a positive offset on the PTR-ToFMS measurements, a negative offset on the
cartridge measurements or both. In contrast, a slope significantly lower than unity (0.58±0.10,
1σ) seems to indicate a negative bias in the response of the PTR-ToFMS measurements, a
positive bias for the cartridge measurements or both.
A calibration issue cannot explain an intercept in the scatter plot but could explain part
of the disagreement observed for the slope. Three potential reasons may lead to a calibration
issue: (i) the generation of unreliable calibration standards, a humidity dependence of (ii) the
PTR-ToFMS response or (iii) the DNPH derivatization. The procedures used to calibrate the
PTR-ToFMS and the HPLC-UV are described in the experimental section. Two different
commercial methylglyoxal solutions were used to generate both the gas-phase standard for the
PTR-ToFMS and the liquid standard for the HPLC-UV. While we cannot rule out an issue with
the MGLY solutions, it seems unlikely that the disagreement between the two techniques is
only due to unreliable MGLY solutions since the good agreement observed on some days (21-
22, 27, 31 July and 1 August) contrasts with the bad agreement observed on other days (17 July,
2-5 August) when MGLY peaks during daytime (200-300 ppt).
As previously mentioned for the PTR-ToFMS calibration, varying the concentration of
MGLY in the range 0.6-11 ppbv with the LCU led to a change in RH. Calculating the calibration
factor at each concentration, i.e. at different RH, from the ratio of the measured normalized
signal-to-the MGLY concentration and plotting it as a function of the m/z 37-to-m/z 19 ratio
(Figure S2) does not indicate a significant water dependence of the PTR-ToFMS response. The
humidity dependence of the DNPH/HPLC-UV method has been recently investigated for some
ketone compounds, including acetone and MEK (Ho et al., 2014b). It was shown that the
collection efficiency is inversely related to relative humidity, with up to 35-80 % of the ketones
being lost for RH values higher than 50% at 22°C. While MGLY exhibits a ketone function it
also exhibits an aldehyde function and it is not clear whether this compound will behave as
simple ketones. The color coding shown in Figure 4 indicates that when higher RH values are
observed (60-100%), lower MGLY concentrations and larger relative differences between the
two techniques are also observed. Figure 5 displays a scatter plot of the difference between the
PTR-ToFMS and DNPH/HPLC-UV measurements and relative humidity, showing a weak
linear correlation with a negative slope. This trend with humidity seems to support that the
collection efficiency of MGLY on DNPH cartridges decreases with RH. It is interesting to note





that a collection efficiency lower than 100%, even at low RH values, may explain lower
concentrations measured by the DNPH/HPLC-UV method, on average, for the overall
campaign.

4        A positive or negative bias in the PTR-ToFMS measurements could be due to an

inadequate peak fitting procedure to separate the signals detected at m/z 73.029 (MGLY) and
m/z 73.065 (MEK + butanal) (see section 2.2). To check whether the peak fitting procedure can
lead to a bias in the measurements, Figure 5 also presents a scatter plot of the difference between
the PTR-ToFMS and DNPH/HPLC-UV measurements and the MEK+butanal concentration
measured by PTR-ToFMS. This scatter plot indicates a very weak correlation ($R^2$ of 0.05),
suggesting that the fitting procedure was able to deconvolute the signals from MGLY and
MEK+butanal. A weaker correlation is even found when the difference is plotted as a function
of butanal, which was measured by DNPH/HPLC-UV (see supplement S3).

13       An ionic water cluster, $(H_2O)_3.H_3O^+$, can also be detected at m/z 73.050. However, as

mentioned previously, the signal from this cluster is recorded during blank measurements and
subtracted from ambient measurements. As a consequence, the detection of $(H_2O)_3.H_3O^+$ at m/z
73 should not impact MGLY measurements reported in this study. Since the abundance of
$(H_2O)_3.H_3O^+$ is highly dependent on the ambient water concentration, relative humidity was
used as a proxy to investigate whether the cluster signal is efficiently recorded in the blank
signal, which was performed hourly to ensure that RH does not change significantly between
two blank measurements. A good correlation ($R^2$=0.45±0.21, from daily analyses) observed
between the blank signal at m/z 73 with the m/z 37-to-m/z 19 ratio (proxy for humidity content),
indicates that the $(H_2O)_3.H_3O^+$ water cluster signal is indeed recorded during blank
measurements.

24       Scatter plots of the difference between PTR-ToFMS and DNPH/HPLC-UV

measurements with $O_3$, acetaldehyde and nopinone were generated (see supplement S3) to
check whether the possible secondary formation of isobaric OVOCs (malondialdehyde, acrylic
acid) in the atmosphere, from the oxidation of ambient VOCs, or in the sampling line from
reaction of $O_3$ with unsaturated compounds adsorbed on surfaces, could lead to a positive bias
in the PTR-ToFMS measurements. The very weak correlations ($R^2$ of 0.02, 0.01 and <0.01 for
$O_3$, acetaldehyde and nopinone, respectively) observed in Figure S3 rule out this possibility.

31       Similar correlation plots were made for m/z 137 (monoterpenes), 139 (Nopinone), 151

(Pinonaldehyde) and 155 (unidentified oxidation product of monoterpenes) measured by PTR-
ToFMS (see supplement S4) to track whether differences observed between both techniques
could be explained by interferences from the fragmentation of larger compounds observed at



significant concentrations during the CharMEx field campaign. Poor correlations were found
($R^2$ < 0.06) suggesting that MGLY measurements were free of interferences from the
fragmentation of compounds measured at these four masses. Nevertheless, we cannot rule out
interferences from the fragmentation of other higher m/z compounds.

5       A closer look at the blank signals measured at m/z 73 shows that this signal correlates

with the total m/z 73 signal on some days (21-22/07, 26-27/07, 01-03/08), with $R^2$ factors
ranging from 0.36-0.56. Lower correlations are observed on other days ($R^2$<0.20). Interestingly,
a scatter plot between the coefficients of determination for the above mentioned correlations
and the daily averaged relative humidity exhibits an anti-correlation (negative slope, $R^2$=0.58)
(see supplement S5). This type of correlation has also been observed by de Gouw et al. (2003),
who explained this behaviour by the sticky nature of MGLY, which could cause a memory
effect in the sampling lines. Different sampling line lengths and characteristics (heated at 50°C
for PTR-ToFMS and not heated for DNPH cartridges, presence of a stainless steel particle filter
and a KI ozone scrubber for DNPH cartridges) could lead to different artefacts related to
adsorption or heterogeneous reaction on line surfaces for the two techniques. It is worth noting
that performing blank measurements every hour and the use of a high flow rate and heated
sampling line likely reduces this artefact for the PTR-ToFMS, while blank measurements for
DNPH cartridges only takes into account passive contamination of the cartridges, without any
artefact from lines considered. It is interesting to note that while the difference between the two
techniques is not correlated to the PTR-ToFMS measurements (see supplement S3), a fair
correlation ($R^2$=0.35) is observed with the cartridge measurements, which may suggest a bias
on the cartridge measurements.

23       While a reasonable agreement is observed between the 2 techniques, a close look at the

correlation between the two measurement sets indicates that the DNPH/HPLC-UV methods
measured lower concentrations than the PTR-ToFMS technique by 18% on average. The above
discussion highlights several potential reasons for this disagreement: (i) calibration standards
of MGLY are difficult to generate for both techniques and require further work to straighten
out this aspect, (ii) the impact of artefacts from sampling lines needs to be further investigated
to evaluate their significance, (iii) the collection efficiency of MGLY in DNPH cartridges needs
to be investigated under ambient sampling conditions to assess whether MGLY is completely
collected and whether there is a humidity dependence.

32       Finally, we cannot exclude that differences observed between PTR-ToFMS and

DNPH/HPLC-UV measurements of MGLY are partly due to differences in sampling sequences
(3h continuous sampling for DNPH/HPLC-UV, 3h sampling minus 3 times ten minutes of blank


measurements for PTR-ToFMS). However, the impact of differences in timescale for the two
techniques should lead to random scatter when the measurements are compared and not to a
systematic difference as observed in this study.
**3.4 MGLY loss rate**

7       Loss rates of MGLY are presented in Figure 6. They were calculated as described in

section 2.4 using PTR-ToFMS measurements since the DNPH/HPLC-UV measurements may
suffer from inlet effects and an overestimated collection efficiency. The total loss rate peaks
during daytime around 14:00 local time at values ranging from 100-350 pptv h$^{-1}$. The calculated
loss rate is almost equally divided into photolysis and oxidation by OH, accounting for 53%
and 47%, respectively, of the average diurnal loss from 10:00 to 19:00 local time.

13       A thorough investigation of the MGLY budget would require calculating the total

MGLY production rate from the oxidation of ambient VOCs for comparison to the total loss
rate presented above. However, as mentioned in the introduction, MGLY is produced during
the oxidation of many VOCs (isoprene, monoterpenes, acetone, aromatics…) at average yields
which are strongly dependent on ambient radical concentrations and NOx as recently reported
for isoprene [Jenkin et al., 2015]. It is also worth noting that calculating MGLY production
rates based on ambient concentrations of precursors and average yield values would only be
robust for first-generation oxidation products since no intermediate species is taken into
account. Taking into account that MGLY is also a second and higher generation oxidation
product in most degradation mechanisms would lead to a delayed formation. Indeed, MGLY
production can take hours in NO rich environments and even days in low NO$_x$ environments
such as this study (Fu et al., 2008). As a consequence, it would be hazardous to try to calculate
local MGLY production rates from the measured VOC precursors.

26       When a gaseous species exhibits a lifetime lower than a few seconds/minutes, such as

radical species or highly photolabile compounds, this species should reach a photostationary
state and chemical production and loss rates should balance each other since transport processes
such as advection and vertical dilution are   too slow to significantly impact the local
concentration of these short-lived species. MGLY exhibits a lifetime of approximately 1-2 h
during daytime, which may be short enough for the photostationary state to hold. In this case,
production rates of MGLY should mimic the loss rate displayed in Figure 6. However,
Washenfelder et al. (2011) showed a breakdown of the photostationary state when applied to
glyoxal, a dicarbonyl compound exhibiting a similar lifetime than MGLY, and as a consequence



the calculated loss rate reported in this study only provides a rough estimation of the local
production rate.

**4 Conclusions and discussion**

To the best of our knowledge, this study presents the first ambient measurements of
methylglyoxal by PTR-ToFMS. This work aims at describing a simple and proper procedure to
perform reliable measurements, relying on (i) the data processing proposed by de Gouw and
Warneke (2007) to account for the impact of ambient humidity on the PTR-ToFMS sensitivity,
(ii) automatized blank measurements performed every hour to suppress potential memory
effects and interferences from water clusters, and (iii) a gaussian peak fitting analysis to
deconvolute the methylglyoxal signal from other compounds exhibiting a similar mass unit but
different exact masses (i.e. butanone and butanal).
The ChArMEx SOP2 field campaign was conducted in an environment characterized
by high biogenic emissions, including isoprene, at the extreme north of the Corsica Island. This
campaign therefore provides a good opportunity to study methylglyoxal measurements, since
this compound is mainly formed via isoprene oxidation. Furthermore, concomitant
measurements of methylglyoxal by PTR-ToFMS and DNPH/HPLC-UV allowed an
intercomparison of these two techniques to test their reliability.
Time series of methylglyoxal measured by both PTR-ToFMS and DNPH/HPLC-UV
revealed concentration levels ranging from 28-365 pptv with a clear diurnal cycle due to the
secondary nature of this compound. The visual comparison of the measured time series shows
a reasonable agreement, with the DNPH/HPLC-UV methods measuring concentrations lower
by 18% on average compared to the PTR-ToFMS technique. A linear regression analysis
performed between the two measurement sets indicates a fair correlation with a determination
coefficient ($R^2$) of 0.48, a slope significantly different than unity (0.58±0.10, 1σ) and a non-
zero intercept (88.3±15.9 pptv, 1σ). Interferences from $(H_2O)_3.H_3O^+$, butanone and butanal can
be excluded for the PTR-ToFMS measurements, validating the procedure used for data
acquisition and analysis. Methylglyoxal formation into sampling lines due to heterogeneous
reactions of $O_3$ with adsorbed organic compounds is also not likely. Potential remaining
uncorrected artefacts from lines on some days for both techniques could be partly responsible
for measurements disagreements and this aspect needs to be further investigated to evaluate its
significance. In addition, this work questions the collection efficiency of MGLY in DNPH
cartridges, recommending to investigate it under ambient sampling conditions to assess whether



all the MGLY is collected and whether humidity dependence exists. Comparisons of PTR-
ToFMS with other existing techniques in the field and/or in atmospheric simulation chambers
would be of interest to identify potential artefacts causing the disagreement observed in this
study. Nevertheless, PTR-ToFMS seems promising for methylglyoxal measurements.
The methylglyoxal loss rate was studied at cape Corsica, revealing that the contributions
of direct photolysis and OH-oxidation were almost similar.
**Acknowledgement**
This study received financial support from Mistrals / ChArMEx programmes, ADEME,
the French environmental ministry, and the CaPPA projects. The CaPPA project (Chemical and
Physical Properties of the Atmosphere) is funded by the French National Research Agency
(ANR) through the PIA (Programme d'Investissement d'Avenir) under contract "ANR-11-
LABX-0005-01" and by the Regional Council Nord-Pas de Calais and the "European Funds for
Regional Economic Development" (FEDER). This research was also funded by the European
Union Seventh Framework Programme under Grant Agreement number 293897, "DEFIVOC"
project and CARBOSOR/Primequal. This study also received funding from the Région Hauts-
de-France, the Ministère de l'Enseignement Supérieur et de la Recherche and the European
Fund for Regional Economic Development through the CLIMIBIO project.
The authors also want to thank Eric Hamonou and François Dulac for logistic
management during the campaign as well as all the participants of the ChArMEx SOP2 field
campaign.

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



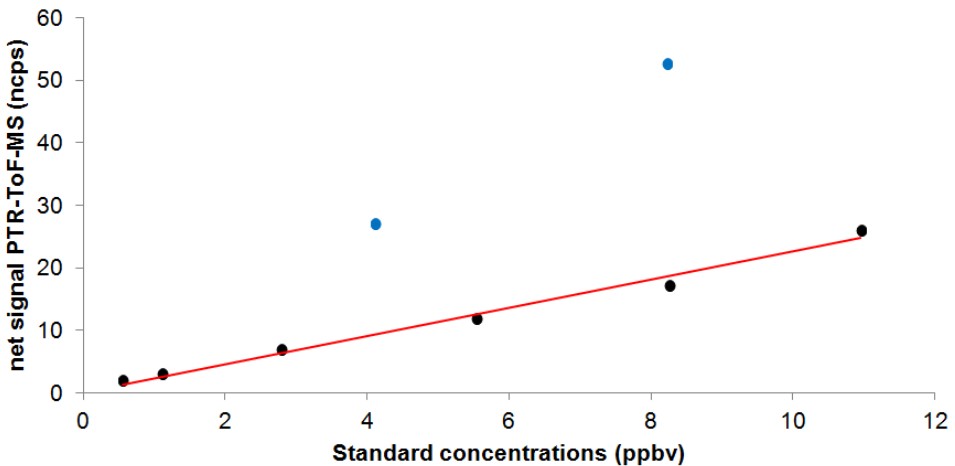

2    Figure 1: MGLY (black circles) and MEK (blue circles) calibration plot for PTR-ToFMS
3    measurements: normalized net signals at m/z 73.029 (MGLY, ncps) and 73.065 (MEK, ncps)
4    vs. generated mixing ratio (ppbv).





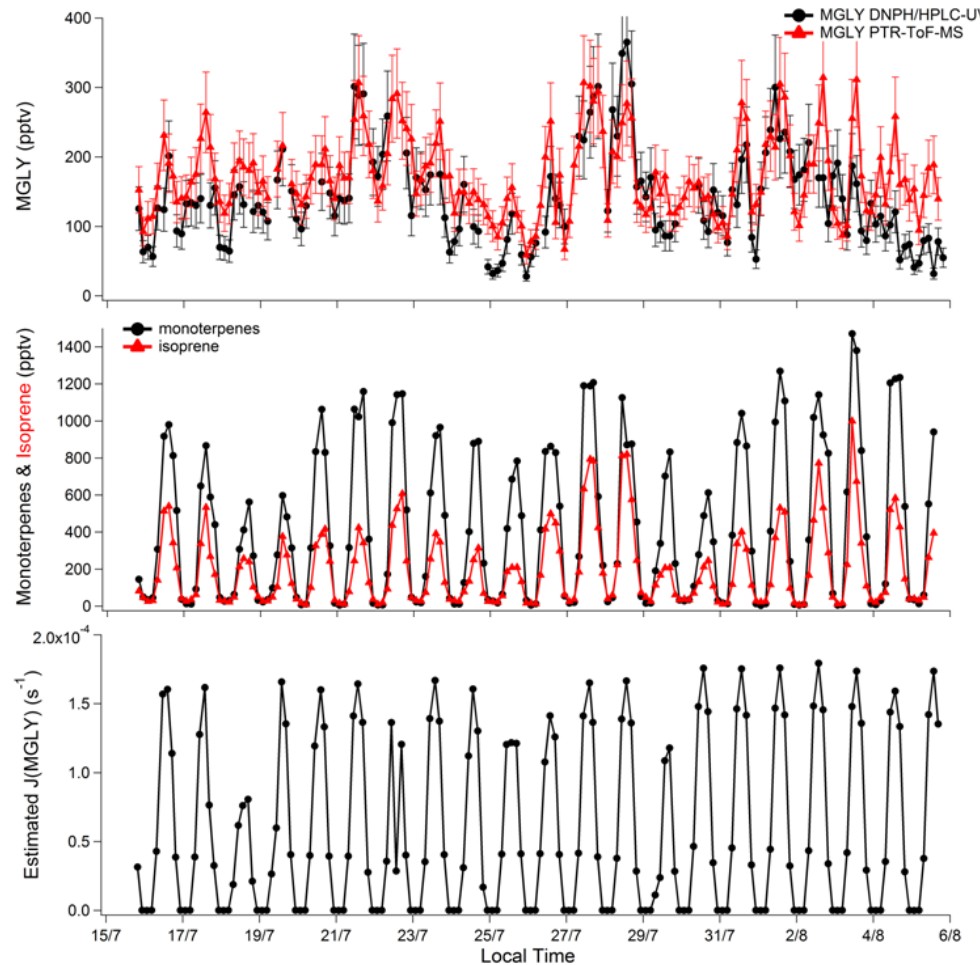

Figure 2 : Time series of MGLY measured by PTR-ToFMS (red) and active sampling on DNPH cartridges (black) (top panel); sum of monoterpenes (black) and isoprene (red) measured by PTR-ToFMS (middle panel); and estimated J(MGLY) (black, bottom panel). Error bars for MGLY measurements (top panel) correspond to systematic errors of 22% and 25% for PTR-ToFMS and DNPH cartridge measurements, respectively.


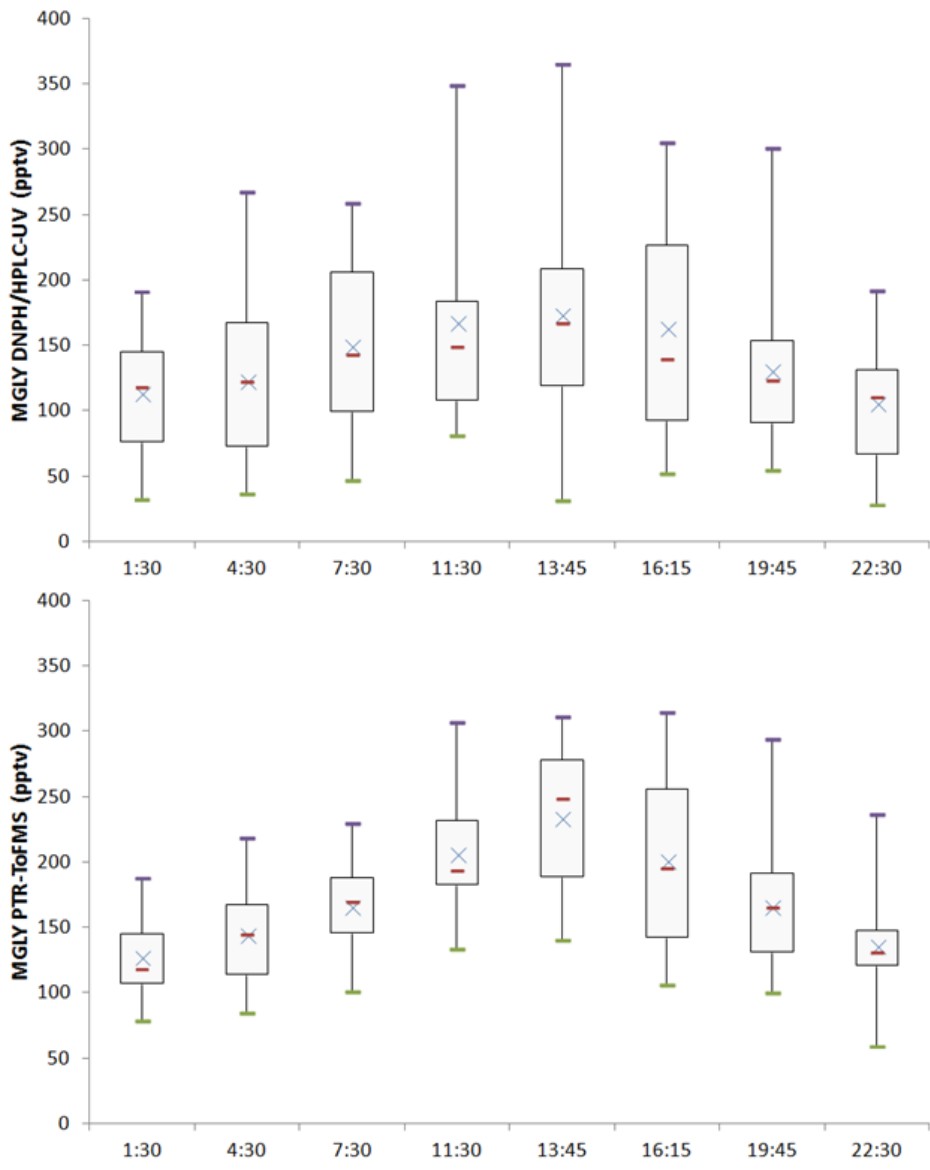

Figure 3: Diurnal profiles (Boxplots) of MGLY measured by both PTR-ToFMS (bottom panel)
and active sampling on DNPH cartridges (top panel) for the campaign average. Purple bars
represent the maxima, green bars the minima, red bars the medians, blue crosses the averages,
and the sides of the boxes: the first (bottom) and the third (top) quartiles.





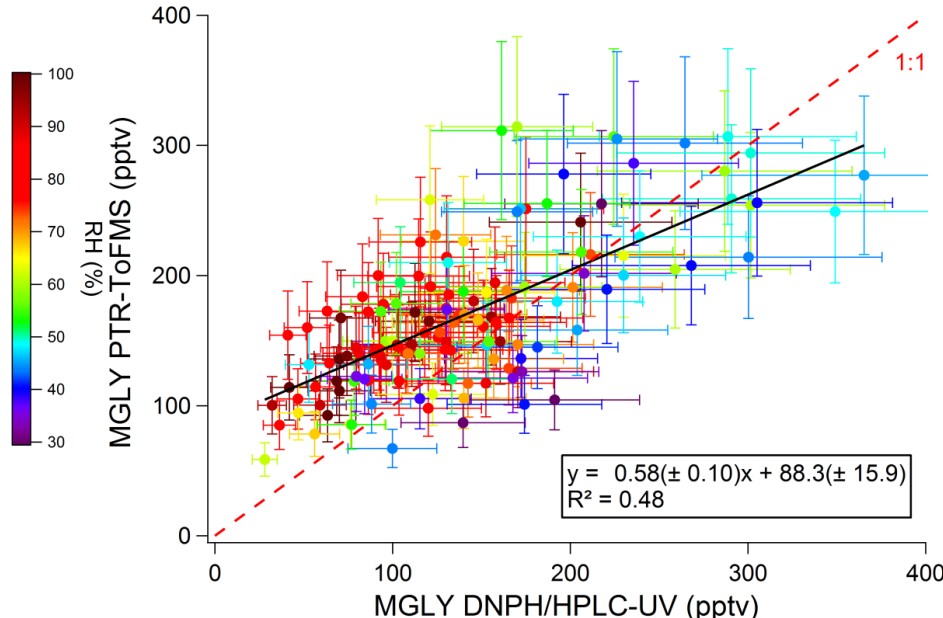

Figure 4: Scatter plot of MGLY concentrations measured by PTR-ToFMS versus
concentrations measured by active sampling on DNPH cartridges. Black line and insert
represent the linear regression. Systematic errors associated to the PTR-ToFMS (22%) and
DNPH cartridge (25%) measurements are accounted for in the regression analysis. The scatter
plot has been color-coded according to the relative humidity.





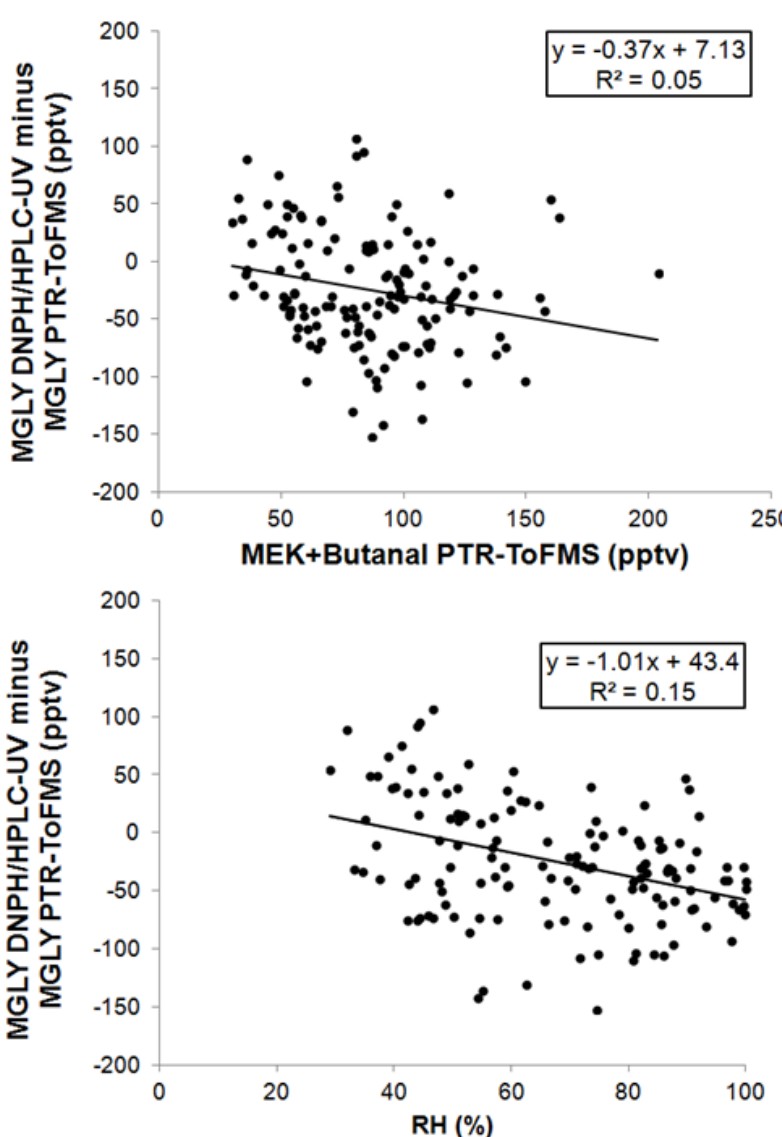

2    Figure 5 : Scatter plots of the difference between the two techniques and MEK+butanal (top
3    panel) or relative humidity (RH, bottom panel).



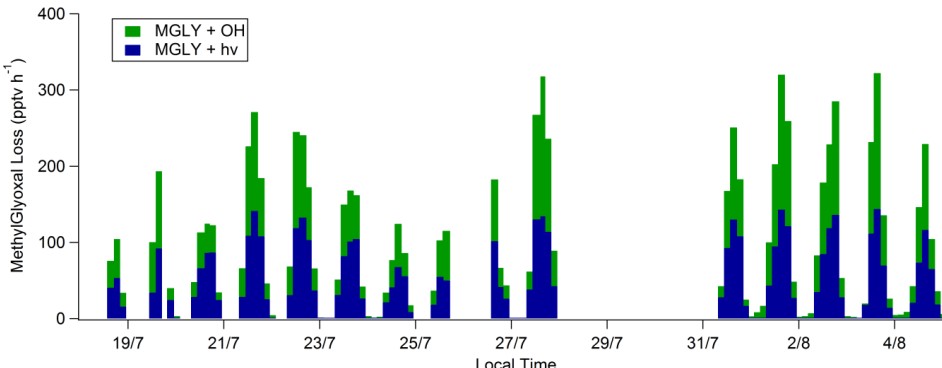

2    Figure 6: Time series of MGLY loss rates (pptv h$^{-1}$) from photolysis and reaction with OH.

