# Peer review of "Field measurements of methylglyoxal using Proton Transfer"

_Atmospheric Measurement Techniques, 2017_

## Referee Comment (RC1) · Anonymous Referee #2 · 20 May 2018

This is an interesting manuscript as it presents concurrent measurements of methylglyoxal by two different measurement techniques during a field campaign at a Mediterranean site impacted by biogenic emissions and discusses potential experimental shortcomings and improvements. The experimental section has been carefully laid out. Here, I only have some minor remarks:

Page 6, L23-25: the authors want to add the information about the mixing ratios and uncertainties of the calibration sources.

[Figure]

Page 7, L13-14: "...using appropriate sensitivity and humidity dependence factors." This is a bit generic. The authors want to elaborate on this and define what they consider "appropriate".

Page 7, L22-23: This combined error accounts about 22%, which is appreciable. It would be interesting to know which of both errors is the major contributor. I assume the authors may be able to state the systematic error of the calibration factor individually and should do it. Apart from that how did the authors estimate the error of the peak fitting procedure?

Page 11, L4: I would not consider a correlation of R2=0.48 to be reasonable. Actually, I find it pretty low for two techniques presumably measuring the same target compound.

The authors perform calculations of the methylglyoxal loss rates and estimate that 53% of methylglyoxal losses are due to photolysis and the remaining losses occur through OH oxidation. As a basis for their calculations the authors use the PTR-ToFMS measurements instead of the DNPH/HPLC-UV measurements as those may be prone to artifacts. As I have learnt from the preceding experimental sections those artifacts predominantly occurred at nighttime. Why would this be relevant for MGLY loss rates, which only occur at daytime? In my opinion, foremost, the higher temporal resolution of the PTR-ToFMS measurements make them more suitable for these calculations than the DNPH/HPC-UV measurements.

Apart from this minor comment, I have some major doubts about the validity of the assumption of photostationary state here. An atmospheric lifetime of 1-2 hrs for MGLY is appreciable. I assume that the measurement site was impacted by land-sea breezes, potentially enhanced by the hilly area. Advective processes on a time scale of 1-2 hrs can definitely not be ruled out.

---

## Referee Comment (RC2) · Anonymous Referee #1 · 13 Jun 2018

Michoud et al. present a series of measurements and acquisition parameters for the use of measuring methylglyoxal (MGLY) in field measurements with Time-of-Flight Proton Transfer Mass Spectrometry (ToF-PTR-MS). They extensively discuss mass spectrometric interferences and work toward accounting for these interferences to provide a highly sensitive (LOD = 22pptv) technique for the detection of MGLY.

I recommend publication with minor revisions as noted below.

General Comments:

[Figure]

The authors present a framework for utilizing ToF-PTR-MS to measure methylglyoxal in the ambient atmosphere. The sampling set up and methodology for the PTR-MS are well developed. What is generally unclear is why the DNPH cartridge sampling method used to compare didn't have a better developed sampling system that would minimize and account for loss mechanisms in the inlet and other bias. If it is known in the literature that MGLY is considered "sticky" for inlets, then steps should be taken to mitigate or quantify the issue, rather than have to use inferences from the field campaign to explain differences later. It would have been a logical next step to apply the calibration procedure used post field campaign with the PTR-MS with the DNPH set up as well to understand and quantify the loss processes of the MGLY in the DNPH set up such as the sampling line and the KI ozone scrubber. As the paper stands the authors have done a good job of sorting out what they can based on the ambient data and the post campaign calibrations with regard to the interference of water clusters and other factors on the MGLY measurements.

In the review of other techniques used to measure MGLY (pages 3-4), the authors rightly point out many of the advantages and disadvantages of other analytical techniques including derivatization and measurement with either GC or HPLC, highly reflective cavity instruments (CE-DOAS and IBBCEAS), and Laser-Induced Phosphorescence. The authors point out the fragile equipment and the need for highly skilled operators as a draw back for these techniques, implying that PTR-MS does not have these as an issue. It could definitely be argued that PTR-MS which comes with a much larger price tag and can certainly be described as temperamental (as any field instrument involving a mass spec and ionization source could be described) certainly need the attention of highly skilled operators and a knowledgeable analyst doing the data work up. The PTR-MS definitely beats out the other techniques in terms of the possible low detection limits, as even in isoprene rich environments such as the LA basin (Washenfelder et al., 2011), while glyoxal is reported by multiple instruments, MGLY is not.

Specific comments:

Page 5 line 21: Methyl glyoxal should also be considered as a 1st generation oxidation product of isoprene oxidation. This was shown mechanistically by Paulot et al. (2009), inferred by Galloway et al. (2011) and measured by Thalman (2013) as a 1.8(0.4)%.

Page 9 line 10: The authors state that the calibration range was higher than the actual ambient range, why not just dilute further to reach the ambient range? Page 10 Line 9: Use "concurrent" rather than "concomitant" which implies that the accompanying of the two measurements was 'naturally occurring'.

Page 11 line 2: As Reviewer 1 has pointed out, an R2 of 0.48 for two techniques measuring the same compound is much worse than expected, though Thalman et al. (2015) reported R2 values for MGLY in mixed compound experiments in the range of 0.65 between the CE-DOAS and SPME techniques.

Page 14 line 34: It is generally considered that methylglyoxal has a much shorter atmospheric lifetime than glyoxal (which is one of the reasons that it is much harder to measure in ambient air), where MGLY has both a larger rate constant for reaction with OH and a 5x larger quantum yield from photolysis (Plum et al., 1983).

References

Galloway, M. M., Huisman, A. J., Yee, L. D., Chan, A. W. H., Loza, C. L., Seinfeld, J. H., and Keutsch, F. N.: Yields of oxidized volatile organic compounds during the OH radical initiated oxidation of isoprene, methyl vinyl ketone, and methacrolein under high-NOx conditions, Atmos. Chem. Phys., 11, 10779-10790, 10.5194/acp-11-10779-2011, 2011.

Paulot, F., Crounse, J. D., Kjaergaard, H. G., Kroll, J. H., Seinfeld, J. H., and Wennberg, P. O.: Isoprene photooxidation: new insights into the production of acids and organic nitrates, Atmos. Chem. Phys., 9, 1479-1501, 10.5194/acp-9-1479-2009, 2009.

Plum, C. N., Sanhueza, E., Atkinson, R., Carter, W. P. L., and Pitts, J. N.: Hydroxyl radical rate constants and photolysis rates of .alpha.-dicarbonyls, Environmental Science & Technology, 17, 479-484, 10.1021/es00114a008, 1983.

Thalman, R., Baeza-Romero, M. T., Ball, S. M., Borrás, E., Daniels, M. J. S., Goodall, I. C. A., Henry, S. B., Karl, T., Keutsch, F. N., Kim, S., Mak, J., Monks, P. S., Muñoz, A., Orlando, J., Peppe, S., Rickard, A. R., Ródenas, M., Sánchez, P., Seco, R., Su, L., Tyndall, G., Vázquez, M., Vera, T., Waxman, E., and Volkamer, R.: Instrument intercomparison of glyoxal, methyl glyoxal and NO2 under simulated atmospheric conditions, Atmos. Meas. Tech., 8, 1835-1862, 10.5194/amt-8-1835-2015, 2015.

Thalman, R. M.: Development of Cavity Enhanced Differential Optical Absorption Spectroscopy (CE-DOAS) and application to laboratory and field measurements of trace gases and aerosols, 3592385, University of Colorado at Boulder, Ann Arbor, 246 pp., 2013.

Washenfelder, R. A., Young, C. J., Brown, S. S., Angevine, W. M., Atlas, E. L., Blake, D. R., Bon, D. M., Cubison, M. J., de Gouw, J. A., Dusanter, S., Flynn, J., Gilman, J. B., Graus, M., Griffith, S., Grossberg, N., Hayes, P. L., Jimenez, J. L., Kuster, W. C., Lefer, B. L., Pollack, I. B., Ryerson, T. B., Stark, H., Stevens, P. S., and Trainer, M. K.: The glyoxal budget and its contribution to organic aerosol for Los Angeles, California, during CalNex 2010, Journal of Geophysical Research: Atmospheres, 116, 10.1029/2011jd016314, 2011.

---

## Author Comment (AC1) · 17 Aug 2018

First, we would like to thank the reviewer 1 for his valuable comments on the manuscript. We did our best to address all the comments and summarized the changes made to the revised manuscript below.

**Response to referee #1:**

*>Page 6, L23-25: the authors want to add the information about the mixing ratios and uncertainties of the calibration sources.*

This information can be found in the supplementary material of the paper written by Michoud et al. (2017). The following sentences have been added in the revised manuscript:
"Information about individual mixing ratios of VOCs in the calibration gases can be found in Michoud et al. (2017; supplementary material S1). Mixing ratios were in the range 0.9-4.5 ppm for the abovementioned species and ranged from 3-15 ppb after dilution with zero air. Uncertainties associated to these mixing ratios range from 5 to 10% ($1\sigma$).

*>Page 7, L13-14: "...using appropriate sensitivity and humidity dependence factors."*
*This is a bit generic. The authors want to elaborate on this and define what they consider "appropriate".*

This statement has been modified in the revised manuscript as follows:
"…using sensitivity and humidity dependence factors determined during calibrations (Xr = 0.5 and 0.49 for MGLY and MEK, respectively)."

*>Page 7, L22-23: This combined error accounts about 22%, which is appreciable. It would be interesting to know which of both errors is the major contributor. I assume the authors may be able to state the systematic error of the calibration factor individually and should do it. Apart from that how did the authors estimate the error of the peak fitting procedure?*

The individual uncertainty for the calibration factor is actually quite large and accounts for 19%. This has been determined as the relative standard deviation of the response factor using all the calibration points shown in Fig S2 and also include the uncertainty of the concentration of the standard generated during calibrations. The error associated to the peak fitting procedure has been estimated to be 10% based on a visual inspection of the Gaussian peak fitting, which is not able to fully represent the two peaks observed at m/z 73 (see Fig S1). The total systematic error has been calculated as the square root of the sum of the squares of these two individual errors, which leads to 22% (rounded up to the superior unit).
This information has been given in the revised manuscript as follow:
"The systematic errors concerned the calibration factor ($R_f$) and the peak fitting procedure and are estimated to be 22% for methylglyoxal (19% and 10% respectively for the individual errors associated to the calibration factor ($R_f$) and the peak fitting procedure)."

*>Page 11, L4: I would not consider a correlation of $R^2=0.48$ to be reasonable. Actually, I find it pretty low for two techniques presumably measuring the same target compound.*

This statement has been modified in the revised manuscript as follow:
"Figure 4 displays a scatter plot of PTR-ToFMS vs. DNPH/HPLC-UV measurements, with a coefficient of determination of 0.48 ($R^2$). A significant intercept of …"

*>The authors perform calculations of the methylglyoxal loss rates and estimate that 53% of methylglyoxal losses are due to photolysis and the remaining losses occur through OH*

*oxidation. As a basis for their calculations the authors use the PTR-ToFMS measurements instead of the DNPH/HPLC-UV measurements as those may be prone to artifacts. As I have learnt from the preceding experimental sections those artifacts predominantly occurred at nighttime. Why would this be relevant for MGLY loss rates, which only occur at daytime? In my opinion, foremost, the higher temporal resolution of the PTR-ToFMS measurements make them more suitable for these calculations than the DNPH/HPC-UV measurements.*

The main discrepancies between both techniques appear at nighttime when MGLY concentrations are at their lowest level, but artifacts for DNPH/HPC-UV measurements due to inlet effects and an overestimation of the collection efficiency might also occur during daytime. This is why we decided to perform MGLY loss rates calculations with the PTR-ToFMS measurements. As mentioned by the reviewer, the higher temporal resolution of the PTR-ToFMS measurements is also an advantage to do these calculations. This last point has been added in the revised manuscript.

References:

Michoud, V., Sciare, J., Sauvage, S., Dusanter, S., Léonardis, T., Gros, V., Kalogridis, C., Zannoni, N., Féron, A., Petit, J.-E., Crenn, V., Baisnée, D., Sarda-Estève, R., Bonnaire, N., Marchand, N., DeWitt, H. L., Pey, J., Colomb, A., Gheusi, F., Szidat, S., Stavroulas, I., Borbon, A., and Locoge, N.: Organic carbon at a remote site of the western Mediterranean Basin: sources and chemistry during the ChArMEx SOP2 field experiment, Atmos. Chem. Phys., 17, 8837-8865, https://doi.org/10.5194/acp-17-8837-2017, 2017.

---

## Author Comment (AC2)

First, we would like to thank the reviewer #2 for his valuable comments on the manuscript. We did our best to address all the comments and summarized the changes made to the revised manuscript below.

**Response to referee #2:**

>The authors present a framework for utilizing ToF-PTR-MS to measure methylglyoxal in the ambient atmosphere. The sampling set up and methodology for the PTR-MS are well developed. What is generally unclear is why the DNPH cartridge sampling method used to compare didn't have a better developed sampling system that would minimize and account for loss mechanisms in the inlet and other bias. If it is known in the literature that MGLY is considered "sticky" for inlets, then steps should be taken to mitigate or quantify the issue, rather than have to use inferences from the field campaign to explain differences later. It would have been a logical next step to apply the calibration procedure used post field campaign with the PTR-MS with the DNPH set up as well to understand and quantify the loss processes of the MGLY in the DNPH set up such as the sampling line and the KI ozone scrubber. As the paper stands the authors have done a good job of sorting out what they can based on the ambient data and the post campaign calibrations with regard to the interference of water clusters and other factors on the MGLY measurements.

The PTR-ToFMS sampling system was optimized to get the best transmission of "very sticky" compounds such as carboxylic acids for this campaign, which also helped for the measurement of other species such as MGLY.

The DNPH method was initially used in this experiment to focus on major carbonyls such as formaldehyde, acetone, acetaldehyde, MEK, MVK and MACR. However MGLY is among the species measurable and thus DNPH method offered this possibility for comparison with the PTRMS. The method has been used with sampling line and ozone scrubber as recommended within TO-11 and EMEP sampling and analysis manual. Indeed, the present study points out some issues to be considered and since then additional studies were performed to improve this method 1) a European study (ENV56/Key-VOCs) tested different materials for sampling lines and sulfinert is now currently used; 2) the collection efficiency of cartridges remains around 100% for aldehydes whatever the relative humidity while a decrease has been observed for ketones above 60% (consistent with Ho et al 2014) leading to the systematic use of two cartridges back to back.. These tests have been done using gaseous standard for carbonyls not including MGLY since it is more challenging to generate mixture with stable concentration of MGLY at a flowrate of 1,5 L min-1. These tests should be performed in a mid-term.

>In the review of other techniques used to measure MGLY (pages 3-4), the authors rightly point out many of the advantages and disadvantages of other analytical techniques including derivatization and measurement with either GC or HPLC, highly reflective cavity instruments (CE-DOAS and IBBCEAS), and Laser-Induced Phosphorescence. The authors point out the fragile equipment and the need for highly skilled operators as a draw back for these techniques, implying that PTR-MS does not have these as an issue. It could definitely be argued that PTR-MS which comes with a much larger price tag and can certainly be described as temperamental (as any field instrument involving a mass spec and ionization source could be described) certainly need the attention of highly skilled operators and a knowledgeable analyst doing the data work up. The PTR-MS definitely beats out the other techniques in terms of the possible low detection limits, as even in isoprene rich environments such as the LA basin (Washenfelder et al., 2011), while glyoxal is reported by multiple instruments, MGLY is not. We agree with the reviewer that operating PTR-ToFMS instruments and processing data also requires highly skilled operators. However, this type of instruments is used more extensively than LIP and cavity instruments during field campaigns. We have added the following statement at the end of the introduction section to clarify these points:

"While PTR-ToFMS instruments also require highly skilled operators and are more expensive than other techniques allowing MGLY measurements, a growing number of research groups is deploying this type of instrumentation during intensive field campaigns, making it of great interest for MGLY measurements. It is expected that PTR-MS should allow reaching a lower LOD than any other techniques reported in the literature so far."

>Page 5 line 21: Methyl glyoxal should also be considered as a 1st generation oxidation product of isoprene oxidation. This was shown mechanistically by Paulot et al. (2009), inferred by Galloway et al. (2011) and measured by Thalman (2013) as a 1.8(0.4)%.

This has been modified in the revised manuscript as follows: "Since MGLY is an oxidation product of isoprene (1st, 2nd & 3rd generations)..."

>Page 9 line 10: The authors state that the calibration range was higher than the actual ambient range, why not just dilute further to reach the ambient range?

We agree with the referee that we should have further diluted the calibration gas to reach, at least, the lowest ambient concentrations. Nevertheless, as stated in the manuscript, we always observed a linear response for the calibrated VOCs on this PTR-ToFMS. There is therefore no indication that the calibration performed for MGLY in this study is not usable for ambient quantification.

>Page 10 Line9: Use "concurrent" rather than "concomitant" which implies that the accompanying of the two measurements was 'naturally occurring'.

This modification has been made in the revised manuscript.

>Page 11 line 2: As Reviewer 1 has pointed out, an R2 of 0.48 for two techniques measuring the same compound is much worse than expected, though Thalman et al. (2015) reported R2 values for MGLY in mixed compound experiments in the range of 0.65 between the CE-DOAS and SPME techniques.

This statement has been modified in the revised manuscript. Please see the answer provided to the first reviewer.

>Page 14 line 34: It is generally considered that methylglyoxal has a much shorter atmospheric lifetime than glyoxal (which is one of the reasons that it is much harder to measure in ambient air), where MGLY has both a larger rate constant for reaction with OH and a 5x larger quantum yield from photolysis (Plum et al., 1983).

Methylglyoxal has indeed a shorter lifetime than glyoxal. However, their global average lifetimes are of the same order of magnitude: 1.6 and 2.9 h for methylglyoxal and glyoxal respectively (Fu et al., 2008). Even if the lifetime of MGLY is shorter by approximately a factor 2 than glyoxal, we cannot rule out a breakdown of the photostationnary state balance and we prefer to be cautious. The manuscript has been modified as follows:

"However, Washenfelder et al. (2011) showed a breakdown of the photostationary state when applied to glyoxal, a dicarbonyl compound exhibiting a lifetime of the same order of magnitude than MGLY, and as a consequence the calculated loss rate reported in this study only provides a rough estimation of the local production rate."

**References:**

Fu, T.-M., Jacob, D. J., Wittrock, F., Burrows, J. P., Vrekoussis, M., and Henze, D. K.: Global budgets of atmospheric glyoxal and methylglyoxal, and implications for formation of secondary organic aerosols, J. Geophys. Res., 113, D15303, doi:10.1029/2007JD009505, 2008.

Ho, S. S. H., Chow, J. C., Watson, J. G., Ip, H. S. S., Ho, K. F., Dai, W. T., and Cao, J.: Biases in ketone measurements using DNPHcoated solid sorbent cartridges, Analytical Methods, 6, 967–974, doi:10.1039/C3AY41636D, 2014

Washenfelder, R. A., Young, C. J., Brown, S. S., Angevine, W. M., Atlas, E. L., Blake, D. R., Bon, D. M., Cubison, M. J., de Gouw, J. A., Dusanter, S., Flynn, J., Gilman, J. B., Graus, M., Griffith, S., Grossberg, N., Hayes, P. L., Jimenez, J. L., Kuster, W. C., Lefer, B. L., Pollack, I. B., Ryerson, T. B., Stark, H., Stevens, P. S., and Trainer, M. K.: The glyoxal budget and its contribution to organic aerosol for Los Angeles, California, during CalNex 2010, J. Geophys. Res., 116, D00V02, doi:10.1029/2011JD016314, 2011.